# Gene flow from Fraxinus cultivars into natural stands of Fraxinus pennsylvanica occurs range-wide, is regionally extensive, and is associated with a loss of allele richness

Everett A. Abhainn[1¤a], Devin L. Shirley[1¤b], Robert K. Stanley[1¤c], Tatum Scarpato[1¤d], Jennifer L. Koch[2], Jeanne Romero-Severson[1] *

1 University of Notre Dame, Department of Biological Sciences, Notre Dame, Indiana, United States of America, 2 USDA Forest Service, Northern Research Station, Delaware, Ohio, United States of America

¤a Current address: Western Ecosystems Technology, Inc., Laramie, Wyoming, United States of America
¤b Current address: Utah Division of Wildlife Resources, Central Regional Office, Springville, Utah, United States of America
¤c Current address: Pacific Northwest National Laboratory, Richland, Washington, United States of America
¤d Current address: Northwestern Medicine Primary Care, Naperville, Illinois, United States of America
* jromeros@nd.edu

## Abstract

In North America, a comparatively small number of *Fraxinus* (ash) cultivars were planted in large numbers in both urban and rural environments across the entire range of *Fraxinus pennsylvanica* Marsh (green ash) over the last 80 years. Undetected cultivar gene flow, if extensive, could significantly lower genetic diversity within populations, suppress differentiation between populations, generate interspecific admixture not driven by long-standing natural processes, and affect the impact of abiotic and biotic threats. In this investigation we generated the first range-wide genetic assessment of *F. pennsylvanica* to detect the extent of cultivar gene flow into natural stands. We used 16 EST-SSR markers to genotype 48 naturally regenerated populations of *F. pennsylvanica* distributed across the native range (1291 trees), 19 *F. pennsylvanica* cultivars, and one *F. americana* L. (white ash) cultivar to detect cultivar propagule dispersal into these populations. We detected first generation cultivar parentage with high confidence in 171 individuals in 34 of the 48 populations and extensive cultivar parentage (23–50%) in eight populations. The incidence of cultivar parentage was negatively associated with allele richness ($R^2$ = 0.151, p = 0.006). The evidence for a locally high frequency of cultivar propagule dispersal and the interspecific admixture in eastern populations will inform *Fraxinus* gene pool conservation strategies and guide the selection of individuals for breeding programs focused on increasing resistance to the emerald ash borer (*Agrilus planipennis* Fairmaire), an existential threat to the *Fraxinus* species of North America.

**Data Availability Statement:** All relevant data are within the manuscript and its Supporting information files.

**Funding:** This work was funded by the National Science Foundation grant IOS1025974 'Comparative Genomics of Environmental Stress Responses in North American Hardwoods' (https://www.nsf.gov/div/index.jsp?div=IOS). JR-S also acknowledges support from USDA-USFS APHIS grant 18-IA-11242316-105 (https://www.fs.usda.gov/), USDA-APHIS grant 20-JV-11242303-050 (https://www.aphis.usda.gov/aphis/home/) and The Tree Fund grant 18-JD-01 (https://treefund.org/). RKS acknowledges support from NIH training grant T32GM075762 (https://cbbi.nd.edu/). JLK acknowledges support from USDA APHIS 18-IA-11242316-105, Michigan Invasive Species Grant Program grant IS18-119 (https://www.michigan.gov/invasives/grants/misgp), the Commonwealth of Pennsylvania Department of Conservation and Natural Resources Bureau of Forestry 18-CO-11242316-014 (https://www.dcnr.pa.gov/Pages/default.aspx) and the U.S. Forest Service Special Technology Development Program grant NA-2017-01.

**Competing interests:** The authors have declared that no competing interests exist.

## Introduction

*F. pennsylvanica*, one of the most wide-ranging of the North American *Fraxinus*, occurs in a multiplicity of forested ecosystems across eastern and central North America [1]. High phenotypic plasticity, cold tolerance, salt tolerance, flood tolerance, ease of clonal propagation, rapid growth, an attractive canopy, and prior to the accidental introduction of the emerald ash borer, few serious insect pests, resulted in over 80 years of widespread and intensive use of this and other closely related *Fraxinus* species for urban landscaping, rural shelterbelts, and riparian buffers [2, 3]. Extensive clonal plantings of a small number of *Fraxinus* cultivars in nearly every city and town across the United States, as well as in rural areas for ecosystem management, could have resulted in gene flow into natural stands, potentially diluting local genetic diversity with propagules from a handful of genotypes. Despite the importance of this species for rural and urban ecosystem management and the severe mortality inflicted by the emerald ash borer (EAB), studies of genetic diversity and population differentiation in the Meliodes, the section of *Fraxinus* native to North America, remain limited to regional provenance tests and local studies using ITS, AFLP, or fewer than 10 genomic microsatellite markers [4–6]. The possibility of extensive gene flow from range wide planting of cultivars remains uninvestigated.

The named cultivars of *F. pennsylvanica* originate from clonal propagation of individual trees selected for desirable characteristics, often directly from a forest setting, or after a few generations of nursery evaluation. All the trees with the same cultivar name are expected to be clonal propagules from a single ortet, the tree from which the clonal propagules originate. In the United States, species purity or known parentage is not required for naming a tree cultivar. As no genetic barrier between *F. pennsylvanica* cultivars and the naturally regenerated trees is likely after so few generations, gene flow from cultivars into naturally regenerated stands is a real possibility.

Propagule dispersal mechanisms in *Fraxinus*, including *F. pennsylvanica*, suggest that gene flow among populations will be high. The fruits of all *Fraxinus* species are samaras, indehiscent winged achenes that enable anemochorous (wind) and hydrochorous (water) seed dispersal. *F. pennsylvanica* samaras float for at least two days and maintain viability after immersion in water for over two weeks [7]. Hydrochorous seed dispersal is the most likely mechanism for the exceptionally rapid spread of *F. pennsylvanica* in Central European floodplain forests where this species is not native; more than 970 km/year in some regions [8]. Given the expected lack of genetic barriers and this well-documented high dispersal ability, *F. pennsylvanica* clonal cultivars could potentially swamp local provenances with nonlocal pollen and seed, especially at the western and northern edges of native range, where populations of apparently local origin are small and widely scattered (S1 Fig) [9]. Although studies of assisted gene flow among conspecific natural populations have attracted interest as an adaptive forest management strategy [10], studies of gene flow from forest tree cultivars into wild conspecifics have primarily focused on the impact of plantation forestry on native gene pools [11–14]. Given the very extensive use of primarily male *F. pennsylvanica* and *F. americana* L. cultivars in urban forestry and the dispersal capacity of both pollen and seed, gene flow into natural populations is certainly possible and may be extensive.

In this investigation we genotyped 48 naturally regenerated populations of *F. pennsylvanica* (1291 trees) and 20 commercial cultivars with the same set of 16 EST-SSR markers to detect possible first-generation progeny from cultivars in the populations we genotyped and assess the impact of such flow on population differentiation and population substructure. We included 10 *F. americana* individuals from a species collection we had genotyped previously to enable detection of misidentification and identify admixed individuals phenotypically

indistinguishable from *F. pennsylvanica* [6, 15]. We detected parentage from cultivars in 34 of the 48 populations and extensive cultivar parentage (23–50%) in eight populations. The incidence of cultivar parentage per site was significantly associated with lower allele richness ($R^2$ = 0.151, p = 0.006). We discuss the implications of this result for landscape genomic studies and gene pool conservation efforts n *Fraxinus* and other native forest tree stands subject to gene flow from native cultivars of nonlocal origin.

## Materials and methods

### Study area and species characteristics

The native range of *F. pennsylvanica* spans more than 23 degrees of latitude and 45 degrees of longitude, extending from Cape Breton Island in the Atlantic Ocean westward to Alberta, Canada and southward to the Gulf Coast of the Eastern United States [16, 17]. In the Great Plains, *F. pennsylvanica* is locally abundant in the Temperate Prairie, West-central Semi-arid Prairie and South-central Semi-arid Prairie ecoregions. In the Eastern Temperate and Northern Forests ecoregions, *F. pennsylvanica* is a significant component of eight forested ecosystems types, including white-red-jack pine in the Great Lakes states and provinces, loblolly-shortleaf pine in the Gulf coastal plains and the Piedmont, oak-pine in the Appalachians, oak-hickory in the more mesophytic areas of the central and eastern United States, oak-gum-cypress in southern bottomlands, elm-ash-cottonwood in seasonally flooded bottomland and aspen-birch on glacial till in cold, moist climates [18]. *F. pennsylvanica* woodlands can also persist in seasonally dry creek beds, upland forests, and seasonally dry urban environments across the entire native range.

Unlike many of the *Fraxinus* species, *F. pennsylvanica* is strictly dioecious (each tree is either male or female), a characteristic that could predispose small local populations to invasion from propagules of clonal cultivars if sex ratios became severely unbalanced. *F. pennsylvanica* is strictly diploid (n = 23). *F. americana*, a closely related species sympatric with *F. pennsylvanica* in many ecoregions in the eastern United States, includes both diploid (n = 23) and polyploid individuals. Many authors assign different species names to the polyploids and comment on the taxonomic uncertainties both within *F. americana* (*sensu lato*) and between *F. americana* (*sensu lato*) and *F. pennsylvanica* [19–23]. This well-known taxonomic uncertainty is why we included *F. americana* individuals in our study, to enable us to detect admixture in our set of cultivars, and therefore in the possible progeny of such cultivars. Our investigation was not designed to assess the range-wide extent of gene flow between *F. pennsylvanica* and *F. americana* or to test hypotheses on the mechanisms driving this gene flow.

### Sample collection design

We collected leaf or twig samples from adult trees (> 10 cm DBH) identified as *F. pennsylvanica* based on taxonomic traits and habitat in 48 naturally regenerated sites (Table 1, Fig 1, S1 Fig). We did not collect samples in southern Michigan, northern Indiana, or northern Ohio, as surviving adult trees were unlikely to be representative of the genetic diversity existing before the emerald ash borer invasion. Populations were identified as all the *F. pennsylvanica* occurring within approximately a square kilometer of naturally regenerated forest or woodland. We sampled at least 30 adult individuals from each site, or all the trees at a site if the census was less than 30. The site locations (Fig 1) and species density (S1 Fig) maps shown are in the USA_Contiguous_Albers_Equal_Area_Conic_USGS_version projection.

**Table 1. Site ID and descriptive statistics for 48 populations.**

| Site ID | N[a] | Latitude | Longitude | $d_{LGM}$ (km)[b] | $F_{ST}$[c] | $A_r$[d] | $P_a$[e] | $N_c$[f] | $I_c$[g] |
|---------|------|----------|-----------|-------------------|-------------|----------|----------|----------|----------|
| SK1 | 24 | 50.79 | -103.89 | 343.7 | 0.104 | 6.6 | 1 | 12 | 0.5 |
| MB2 | 26 | 50.56 | -96.63 | 410.7 | 0.065 | 8 | 3 | 3 | 0.115 |
| AB1 | 27 | 50.02 | -110.75 | 216.5 | 0.111 | 6.4 | 1 | 11 | 0.407 |
| MB1 | 27 | 49.69 | -95.28 | 432.4 | 0.11 | 6.6 | 1 | 5 | 0.185 |
| QB1 | 35 | 48.68 | -71.90 | 227.1 | 0.111 | 6.3 | 2 | 9 | 0.257 |
| MN2 | 28 | 47.63 | -94.61 | 250.7 | 0.088 | 7.6 | 2 | 3 | 0.107 |
| MN1 | 32 | 47.07 | -93.93 | 238.7 | 0.085 | 7.1 | 0 | 7 | 0.219 |
| ND1 | 27 | 46.83 | -100.83 | -106.1 | 0.12 | 6.4 | 0 | 9 | 0.333 |
| ND2 | 27 | 46.80 | -100.81 | -109.6 | 0.099 | 6.3 | 1 | 13 | 0.481 |
| NS3 | 32 | 46.72 | -60.93 | 128.6 | 0.217 | 6.4 | 0 | 1 | 0.031 |
| MT1 | 35 | 46.37 | -105.06 | -49.4 | 0.119 | 6.4 | 2 | 11 | 0.314 |
| MT2 | 24 | 46.34 | -105.07 | -45.5 | 0.158 | 5.7 | 1 | 9 | 0.375 |
| NB2 | 25 | 45.95 | -66.88 | 233.6 | 0.216 | 6.3 | 0 | 1 | 0.04 |
| NB1 | 29 | 45.87 | -66.51 | 230.5 | 0.138 | 6.3 | 2 | 4 | 0.138 |
| SD2 | 29 | 45.74 | -97.23 | 75.7 | 0.074 | 6.7 | 0 | 7 | 0.241 |
| SD1 | 25 | 45.65 | -97.13 | 90.5 | 0.099 | 7 | 0 | 5 | 0.2 |
| QB3 | 30 | 45.53 | -76.02 | 590.8 | 0.082 | 6.9 | 1 | 8 | 0.267 |
| WI2 | 27 | 45.31 | -88.56 | 171 | 0.103 | 8.1 | 4 | 0 | 0 |
| WI1 | 27 | 45.13 | -88.44 | 174 | 0.057 | 7.9 | 0 | 4 | 0.148 |
| VT1 | 27 | 44.97 | -73.27 | 276.2 | 0.111 | 5.9 | 1 | 6 | 0.222 |
| ON1 | 25 | 44.84 | -75.30 | 191.5 | 0.032 | 9.2 | 6 | 1 | 0.04 |
| NS2 | 24 | 44.90 | -63.84 | 124.8 | 0.081 | 7.3 | 3 | 0 | 0 |
| WY1 | 12 | 44.77 | -104.67 | -205.9 | 0.112 | 5.1 | 0 | 1 | 0.083 |
| VT2 | 24 | 44.74 | -73.26 | 293.8 | 0.077 | 6.6 | 0 | 0 | 0 |
| WY2 | 28 | 44.53 | -104.08 | -235.6 | 0.092 | 6.8 | 2 | 5 | 0.179 |
| IA1 | 28 | 42.04 | -93.60 | -9.8 | 0.128 | 6.3 | 4 | 0 | 0 |
| IA2 | 31 | 42.04 | -93.64 | -8.8 | 0.088 | 6.4 | 0 | 10 | 0.323 |
| NE2 | 27 | 41.30 | -102.12 | -522 | 0.131 | 6.1 | 0 | 6 | 0.222 |
| NE1 | 30 | 41.24 | -101.67 | -551.4 | 0.126 | 5.8 | 1 | 4 | 0.133 |
| IL2 | 29 | 40.96 | -89.43 | -19.3 | 0.084 | 6.9 | 0 | 5 | 0.172 |
| IL1 | 30 | 40.44 | -89.96 | -79.1 | 0.099 | 7.1 | 1 | 1 | 0.033 |
| OH1 | 30 | 40.12 | -82.97 | -13.6 | 0.069 | 9.1 | 3 | 0 | 0 |
| OH2 | 24 | 40.01 | -82.84 | -7.6 | 0.093 | 7.7 | 1 | 1 | 0.042 |
| MO2 | 30 | 39.01 | -92.77 | -341.4 | 0.072 | 7.8 | 4 | 0 | 0 |
| MO1 | 29 | 38.87 | -92.30 | -323.4 | 0.152 | 7.8 | 3 | 1 | 0.034 |
| VA2 | 27 | 38.83 | -77.50 | -363.6 | 0.071 | 8.7 | 4 | 0 | 0 |
| KS1 | 29 | 38.72 | -94.96 | -359.1 | 0.087 | 7.1 | 4 | 4 | 0.138 |
| VA1 | 29 | 37.62 | -78.21 | -392.4 | 0.069 | 8.4 | 1 | 0 | 0 |
| TN1 | 29 | 35.69 | -83.50 | -450.6 | 0.238 | 6 | 2 | 0 | 0 |
| OK1 | 10 | 35.40 | -95.59 | -700.2 | 0.164 | 4.3 | 0 | 3 | 0.3 |
| SC2 | 17 | 33.14 | -79.82 | -718.7 | 0.155 | 5.6 | 2 | 0 | 0 |
| SC1 | 28 | 32.95 | -79.75 | -738.3 | 0.115 | 7.2 | 2 | 0 | 0 |
| MS1 | 26 | 32.82 | -90.81 | -851.9 | 0.154 | 6 | 0 | 1 | 0.038 |
| MS2 | 27 | 32.71 | -90.79 | -840.7 | 0.112 | 7.6 | 2 | 0 | 0 |
| TX1 | 21 | 30.56 | -95.75 | -1171.6 | 0.1 | 6.6 | 0 | 0 | 0 |
| LA1 | 29 | 30.41 | -91.67 | -1092.7 | 0.094 | 8 | 3 | 1 | 0.034 |
| LA2 | 24 | 30.39 | -89.73 | -1075.5 | 0.146 | 6 | 0 | 0 | 0 |

*(Continued)*

**Table 1.** (Continued)

| Site ID | N[a] | Latitude | Longitude | $d_{LGM}$ (km)[b] | $F_{ST}$[c] | $A_r$[d] | $P_a$[e] | $N_c$[f] | $I_c$[g] |
|---------|------|----------|-----------|----------|------|------|------|------|------|
| TX2 | 31 | 29.17 | -95.50 | -1305.4 | 0.142 | 6.7 | 0 | 0 | 0 |

[a]Number of individuals

[b]Distance from the LGM (negative outside of LGM, positive within LGM),

[c]Population-specific $F_{ST}$,

[d]Allele richness,

[e]Private alleles,

[f]Number of cultivar parents detected,

[g]Incidence of cultivar parents

## Cultivars

Cultivars were from the collection at the Minnesota Landscape Arboretum (Chaska, MN). A cultivar checklist made in 1983 listed 23 valid *F. pennsylvanica* cultivars [24]. We initially evaluated 20 named cultivars (S1 Table), 12 of which were on the 1983 list and eight of which were named after this date. 'Lednaw Aerial' was an exact match to 'Summit'. We discovered later that 'Lednaw Aerial' is a bud sport (a somatic mutation) of 'Summit', which explained their identical genetic profiles. Fourteen cultivars from northern sources dominate this collection: five from Minnesota, five from North Dakota, two from Iowa, one from Alberta and one from Wisconsin. The 10 *F. americana* included as comparators were identified as *F. americana* based on site characteristics, morphological characteristics (primarily the shape of bud scar and dormant bud), AFLP sequences, and ITS sequences [25]. 'Cimmzam Cimmaron' was included in this group because it was thought be *F. americana* with some *F. pennsylvanica* admixture [23, 26].

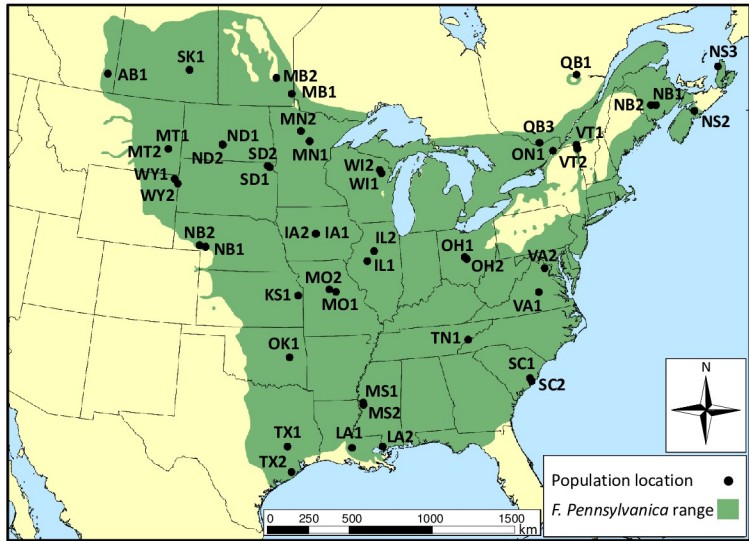

**Fig 1. Site locations and identifiers for 48 *Fraxinus* populations relative to the native range of *F. pennsylvanica*.**

## DNA extraction, PCR, and genotyping

DNA extraction techniques and PCR protocols were as previously described [15]. We used 16 EST-SSR markers developed previously for *Fraxinus* to identify genetic variation within and among populations and closely related species [15, 27]. All 16 of these EST-SSR markers were selected for consistent amplifiability and polymorphism in 187 *F. pennsylvanica* and eight *F. americana*. Fifteen were chosen from the *F. pennsylvanica* transcriptome and one from *F. excelsior* (GenBank: FR639289.1). Amplicon length was measured with an ABI 3730xl capillary electrophoresis device (Applied Biosystems, Foster City, CA) and the resulting genotypes were scored with GENEMAPPER version 4.1 (Applied Biosystems).

## Statistical analyses

Parentage analysis was done with CERVUS version 3.0.7 The mean polymorphic information content (PIC) of the data used to assign parentage was 0.7418 and the combined exclusion probability with one parent known exceeded 0.999. These values are similar to a parentage analysis of cultivated Pacific oysters in which the authors found that the combined exclusion power of 12 microsatellites for identification of one parent was 1 (100% correct identification of the true parent) while more than 50 SNPs were required for the same result [28]. The settings for parentage analysis simulations for the cultivars were 10,000 offspring, 0.05 proportion of candidate pollen parents sampled, 0.05 proportion of loci mistyped, and a minimum of 12 loci typed, yielding a critical LOD score of 2.14 for a strictly confident parent/offspring pairs (95% confidence).

We employed GENALEX version 6.51b2 [29, 30] to detect private alleles, assess isolation-by-distance, and generate pairwise *F*-statistics [31]. We used the Bayesian approach implemented in GESTE 2.0 to estimate population-specific differentiation, permitting evaluation of spatial factors as predictor variables for the genetic distinctness of each population relative to all the other populations [32]. Analysis parameters used were 100 pilot runs, a sample size of 10,000, thinning interval of 20, and a burn-in of 50,000. Evaluation of spatial factors and cultivar incidence as predictor variables for allele richness was done using simple linear regression models as implemented in the version of Excel included with Windows 10 Pro.

Population structure and admixture were estimated with STRUCTURE version 2.3.4 [33] with 50,000 MCMC burn-in iterations followed by 100,000 MCMC iterations. All of the data, (populations, cultivars, and the *F. americana* comparators) were included. We initially tested *K* (the number of proposed genetic groups) from two to 20, with 10 replicates for each *K*. The *K* at which the data were most likely was inferred using the Evanno method as implemented in STRUCTURE HARVESTER [34, 35]. Individuals were counted as admixed if the estimated proportion of membership in a single group was < 0.85. A value > 0.85 falls within average 95% confidence interval for unadmixed membership for the data in this analysis.

## Results

### Cultivars and cultivar parentage

The mean polymorphic information content (PIC) of the data used to assign parentage was 0.7418 and the combined exclusion probability exceeded 0.999. Allowing for one mismatched allele in the comparison and a minimum of 12 loci typed, none of the trees genotyped from natural stands matched individual cultivar genotypes. Parentage analysis detected 171 high confidence (95%) parent-offspring matches in which one of the parents was identified as one of the 19 cultivars (Table 2). Widespread planting of clonally propagated landscaping cultivars in locations far distant from the reported origin is revealed in the distribution of some cultivar

**Table 2. Number (N) and incidence (I) of cultivar parentage by site ID and cultivar[a].**

| Site ID | BRG | CC | CP | EM | HF | HS | JWL | JL | KIN | LPD | MAN | MS | NWP | PAT | RPS | SD | SUM | WDC | N | I |
|---|---|---|---|---|---|---|---|---|---|---|---|---|---|---|---|---|---|---|---|---|
| SK1 | 1 | | | | | 2 | | | | 1 | | | 3 | | 1 | | 3 | 1 | 12 | 0.5 |
| MB2 | 1 | | | 1 | | | | | | | | 1 | | | | | | | 3 | 0.12 |
| AB1 | 1 | | | | | | 1 | | | 2 | | 2 | 1 | 1 | | | 3 | | 11 | 0.42 |
| MB1 | | | | | | 1 | | | | | | | | 4 | | | | | 5 | 0.19 |
| QB1 | 1 | | 1 | | | | | | | 4 | | | | | 2 | | | 1 | 9 | 0.32 |
| MN2 | | | | | | | | | | | | | | | | | | 3 | 3 | 0.1 |
| MN1 | 1 | | | | | 1 | 1 | | | 1 | | | 1 | | | | | 2 | 7 | 0.23 |
| ND1 | | | 1 | | 2 | 1 | | | | 4 | | | | | 1 | | | | 9 | 0.29 |
| ND2 | 6 | | | | 1 | | 2 | 1 | | 2 | | | | | | | | 1 | 13 | 0.41 |
| NS3 | | 1 | | | | | | | | | | | | | | | | | 1 | 0.03 |
| MT1 | 1 | | | | | 1 | | | | 2 | | 2 | | | 4 | 1 | | | 11 | 0.32 |
| MT2 | | | | | | | 1 | | | 3 | | 2 | 2 | | | | 1 | | 9 | 0.26 |
| NB2 | | 1 | | | | | | | | | | | | | | | | | 1 | 0.03 |
| NB1 | | | | | | 1 | | | | | | | | | 2 | | 1 | | 4 | 0.11 |
| SD2 | | | | | | 3 | | | | | | 1 | | | 2 | | 1 | | 7 | 0.18 |
| SD1 | | | | | | | 1 | | | 1 | | | 2 | | 1 | | | | 5 | 0.13 |
| QB3 | 1 | | | | | | | 2 | | 1 | | | | | 1 | | 3 | | 8 | 0.2 |
| WI1 | | | | | | | | 1 | | | | | 1 | | 1 | | 1 | | 4 | 0.1 |
| VT1 | 1 | | | 1 | 1 | | | 1 | | 1 | | | | | | | 1 | | 6 | 0.14 |
| ON1 | | 1 | | | | | | | | | | | | | | | | | 1 | 0.02 |
| WY1 | | | | | | | | | | | | | | | | | 1 | | 1 | 0.02 |
| WY2 | 1 | | | 1 | | | | 1 | | 1 | | | | | | | 1 | | 5 | 0.11 |
| IA2 | | | | | | | 1 | 1 | 1 | | 1 | 4 | 1 | | | | | 1 | 10 | 0.22 |
| NE2 | | 1 | | | | 1 | | | | 2 | | | | | 2 | | | | 6 | 0.13 |
| NE1 | | | | | | 2 | | | | | | | | | 1 | | | 1 | 4 | 0.08 |
| IL2 | | | | | | 2 | 1 | | | | | 2 | | | | | | | 5 | 0.1 |
| IL1 | | | | 1 | | | | | | | | | | | | | | | 1 | 0.02 |
| OH2 | | 1 | | | | | | | | | | | | | | | | | 1 | 0.02 |
| MO1 | | | | | | | | | | | 1 | | | | | | | | 1 | 0.02 |
| KS1 | | | 1 | | 2 | | | | | | | 1 | | | | | | | 4 | 0.08 |
| OK1 | | | 2 | | | | | | 1 | | | | | | | | | | 3 | 0.06 |
| MS1 | | | | 1 | | | | | | | | | | | | | | | 1 | 0.02 |
| LA1 | | | 1 | | | | | | | | | | | | | | | | 1 | 0.02 |
| N | 15 | 5 | 5 | 6 | 7 | 15 | 8 | 6 | 2 | 25 | 2 | 15 | 11 | 5 | 18 | 1 | 17 | 9 | | |

[a]The 15 locations in which no cultivars were detected not shown. The cultivar 'Hollywood,' not detected as a parent in any location, not shown.

offspring (Table 2). 'Leeds Prairie Dome', reported as originating in North Dakota, was identified as the parent of 25 progeny distributed across seven states and three provinces, from southern Saskatchewan to northern Quebec and south to western Nebraska. 'Emerald', reported as originating from Arlington, Nebraska was identified as the parent of six progeny, from Manitoba to Mississippi. 'Cimmzam Cimmaron', was identified as the parent of one tree in each of five sites (Table 2). Only 'Hollywood' had no parent-offspring matches.

Cultivar parentage was not detected in 15 of the 48 sites (WI2, NS2, VT2, IA1, OH1, MO2, VA2, VA1, TN1, SC1, SC2, MS2, TX1, TX2, LA2). Cultivar parentage comprised 23–50% of the individuals tested at eight sites, seven located in the northwestern region of the range (SK1, AB1, MN1, ND1, ND2, MT1 and MT2) and one (QB1) in the northeast (Table 2). Twelve

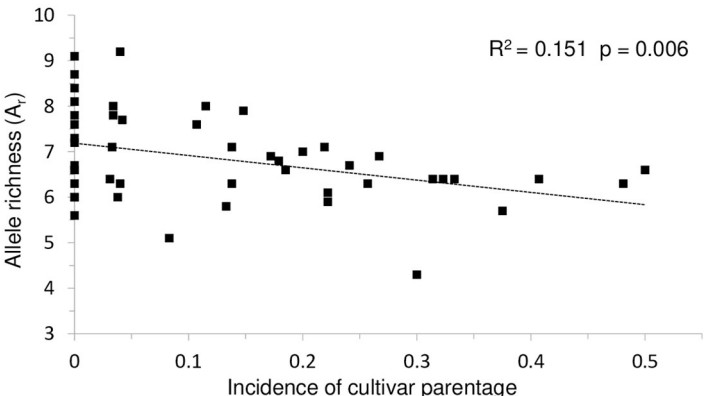

**Fig 2. Higher incidence of cultivar parentage associated with lower allele richness.**

individuals of the 24 sampled at the Saskatchewan site had parentage from seven different cultivars. As most of the cultivars examined are males (S1 Table), offspring likely resulted from pollen flow into the population. We infer that the offspring of the female trees 'Jewel' and 'Mandan' are the result of pollen flow from naturally regenerated trees followed by seed dispersal back into the naturally regenerated stand.

## Association between incidence of cultivar parentage, spatial factors, and allele richness

The incidence of cultivar parentage was higher at higher latitudes ($R^2 = 0.325$, $p < 0.001$) and higher at the more negative longitudes, i.e. towards the west ($R^2 = 0.237$, $p < 0.001$). There was no evidence for an association of latitude with allele richness ($R^2 < 0.001$, $p = 0.98$) and a non-significant association of longitude with allele richness ($R^2 = 0.054$, $p = 0.11$).

A high incidence of cultivar parentage was significantly associated with lower allele richness ($R^2 = 0.151$, $p = 0.006$, Fig 2, S2 Table). If the 15 sites for which no cultivar parentage was detected are removed from the analysis, the significance of the association remains ($R^2 = 0.158$, $p = 0.021$), demonstrating that the association is not an artifact of the 15 values of zero for sites in which no cultivar parentage was detected.

## Population differentiation

Molecular variance within populations accounted for 70% of the total, followed by the variance within individuals (16%) and last, the variance among populations (14%). Pairwise $F_{ST}$ values ranged from 0.013 (SK1-AB1) to 0.204 (ON1-MO2). We found a significant signature of isolation-by-distance ($R^2 = 0.15$, $p = 0.001$). Population-specific $F_{ST}$ estimates, a measure of the differentiation of each population from all of the others, ranged from 0.03 in Ontario 1 to 0.24 in Tennessee 1. Posterior model probabilities for an association of population-specific $F_{ST}$ estimates with latitude ($P = 0.052$) and longitude ($P = 0.023$) did not suggest latitudinal or longitudinal gradients. The distance from the last glacial maximum (dLGM) had the highest posterior probability ($P = 0.111$) but this value was lower than the minimum (0.15) recommended [32]. The interpretation of this data is not straightforward, given the evidence for species admixture, which may or may not be significantly influenced by a human-mediated process and cultivar gene flow, which most certainly is.

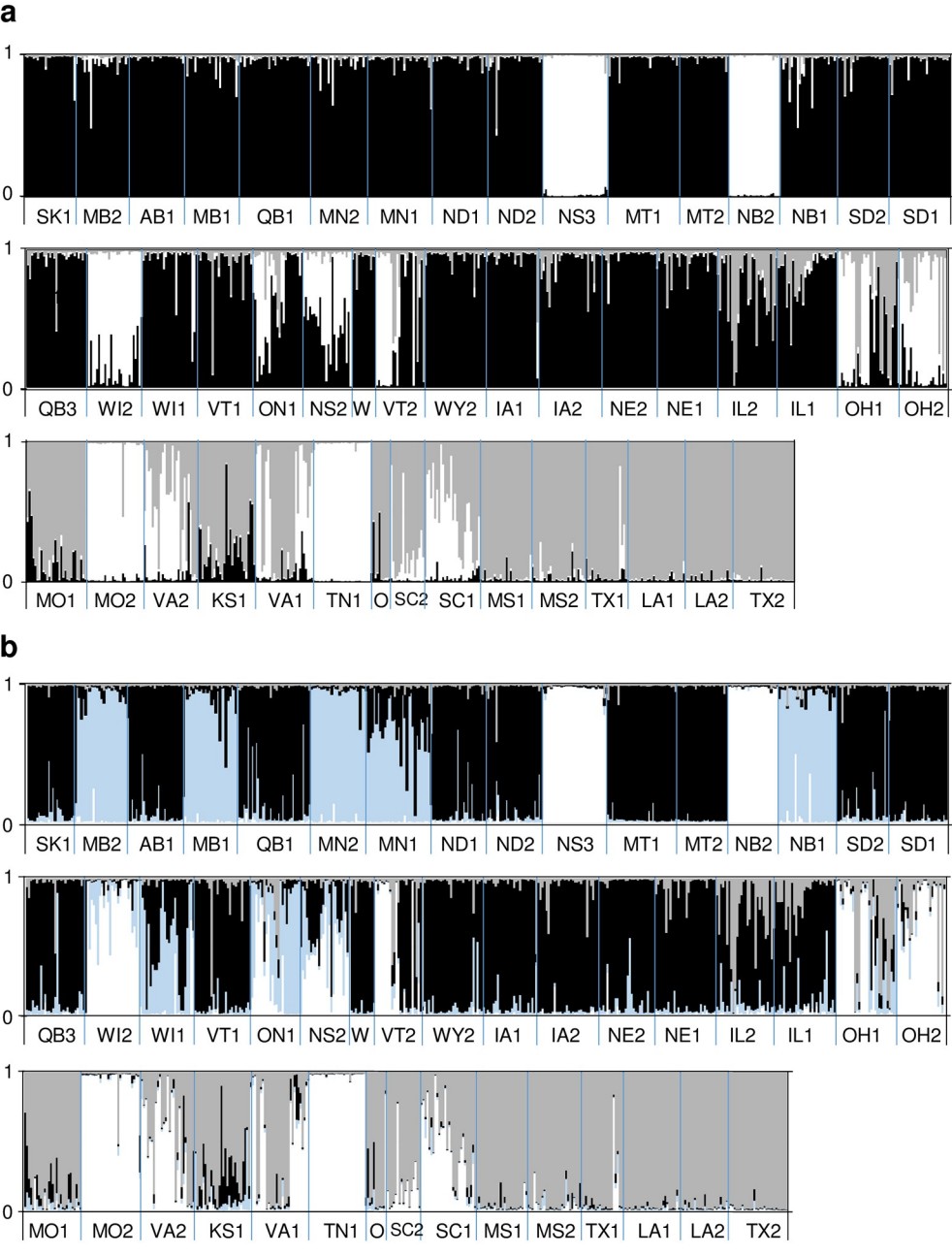

**Fig 3. Genetic groups and admixture inferred for 48 *Fraxinus* populations.** Panel a) Genetic groups and admixture inferred at *K* = 3. Panel b) Genetic groups and admixture at *K* = 4.

## Genetic structure and population admixture

The data were most likely at *K* = 3 genetic groups when assessed with STRUCTURE (Fig 3a, S2 Fig). The analyses were performed on all 48 populations, 19 cultivars and 10 *F. americana* comparators as a single data set, but the results are shown by populations only (Fig 3) and by cultivars and comparators only (S2 Fig) for clarity. Two groups mapped on two broad geographical regions, the northwestern and the southern part of the native range. The

northwestern group lies, for the most part, in the Great Plains region of the United States and Canada. The third group has a geographically dispersed distribution, primarily in the eastern part of the native range. When visualized at *K* = 4, the Northwestern group splits into two groups, Northwestern and Northern (Fig 3b), while the remaining two groups remain largely unchanged. A high frequency of individuals with admixture from all three groups occurred in the Ohio and Virginia sites and to a lesser extent in the Ontario site. Individuals in three sites (NS3, NB3 and TN1) cluster in group three, with minimal admixture.

### *F. americana* and *F. americana* admixture

Seven of the 10 presumed *F. americana* comparators were unadmixed members of the third group (S2 Fig). While this does not prove that the third group is *F. americana*, it is suggestive. At K = 3, Trees FA_7 and FA_9 are highly admixed and tree FA_3 admixed with the *F. pennsylvanica* group. At K = 4, FA_3 and FA_7 group with what we designate as the 'northern group' of *F. pennsylvanica* while tree FA_3 is admixed with all three. This result suggests that identification by morphology and short DNA sequences (AFLP and ITS) does not necessarily detect admixture or even species in some cases, despite expert close examination. If the third group is in fact, *F. americana*, then the investigators in this study, guided by local experts using morphological and habitat criteria, provide an excellent example of the difficulty with identification in field situations in locations where F. pennsylvanica and *F. americana* are sympatric.

## Discussion

We designed our range-wide study to assess the extent of gene flow from cultivars into naturally regenerated stands of *F. pennsylvanica*. The possibility of such gene flow seemed likely given the propagule dispersal capacity of *Fraxinus* and the extensive planting of *F. pennsylvanica and F. americana* in urban and agricultural environments across the native range for both species over the last 80 years. We expected to detect evidence of gene flow from cultivars into local stands in the northwest region of the natural range where populations are sparsely distributed and where most of the cultivars in this study originated, but we did not expect the frequency of cultivar parentage we detected. We expected to see low differentiation among *F. pennsylvanica* populations across broad regional scales, given the species high dispersal capacity, high phenotypic plasticity, and high tolerance to abiotic stress. We included a small group of *F. americana* comparators in our study to permit us to detect misidentification.

### Gene flow from landscaping cultivars in naturally regenerated populations

**The propagule dispersal capacity of *Fraxinus* species.** As this investigation is the first range-wide study of *F. pennsylvanica* using genetic markers and our question focused on evidence for gene flow from cultivars, our expectation for what we might find range-wide was based on evidence from other *Fraxinus* species and some informed speculation. Rapid hydrochorous seed dispersal, coupled with anemochorous seed and pollen dispersal, could potentially minimize local differentiation in native stands while maintaining high standing genetic variation across broad regional scales. Populations of *F. excelsior* (European ash) in Britain and France show minimal differentiation ($F_{st}$ = 0.025), suggesting extensive propagule exchange across broad geographical regions [36]. A similar study in Ireland also detected very low differentiation, little indication of inbreeding and high genetic diversity throughout the island [37]. However, a larger population study across most of the range of *F. excelsior*, while supporting regional panmixia among British, western European, and central European populations, found strong genetic differentiation between the three Swedish populations and the southeast European populations [38]. A more detailed study of far northern range edge

populations revealed high population differentiation and loss of genetic diversity relative to the more southern populations, the expected signal of postglacial colonization [39]. Based on these data, it might be reasonable to assume that the range-wide population dynamics of *F. pennsylvanica* would be similar. However, the native range of *F. pennsylvanica* lacks the altitudinal and coastal heterogeneity present within the range of *F. excelsior*. Patterns of glacial advance and retreat in North America were different than those in Europe and Neolithic human impacts on the landscape differed substantially from those in Europe [40–42]. The absence of impassable geographical barriers in the central and eastern United States and Canada and the high dispersal capacity of *Fraxinus* suggests that high genetic diversity and low genetic differentiation may be present across most of the range. Alternatively, the high climate contrasts between the Great Plains, the Gulf coasts and the Atlantic coasts may have resulted in regional differentiation as a result of adaptive variation.

**The lack of evidence for spatial gradients.** Intensive and widespread planting of ash cultivars for decades may have contributed to the lack of evidence for spatial gradients in population differentiation in the populations we examined. As our investigation of gene flow from landscaping clones does not include all of the named cultivars released in the last 40 years, our results may be an underestimate of the actual frequency of cultivar parentage in naturally regenerated stands and the geographic extent of cultivar gene flow into these stands. The potential impact of gene flow from cultivars and from *F. americana* on the population dynamics of *F. pennsylvanica* cannot be disentangled from the climatic and geographic factors that may have influenced spatial gradients across the native range of *F. pennsylvanica* before widespread planting of cultivars, at least within the limits of the detection afforded with 16 EST-SSR markers.

The Fraxinus section Meliodes occurs only in North America and includes *F. pennsylvanica*, *F. americana*, *F. profunda* (Bush) Bush, *F. caroliniana* Mill. and others. Landscape genomic studies using high-density marker arrays designed to function well across all of the Meliodes and the cultivars derived from these species could more accurately detect the extent of cultivar gene flow and the degree of interspecific admixture among these species. These studies are essential for an accurate characterization of the Meliodes pangenome, including the extent of interspecific gene flow and identification of genetic networks that contribute to stress resistance, especially resistance to the emerald ash borer.

**A conservation conundrum.** In the northern Great Plains region of the United States and Canada, where the northwestern populations are located, *F. pennsylvanica* stands are small and widely scattered. Our data suggests that some of these populations may consist primarily of the descendants of propagules dispersed from cultivars or the improved germplasm planted for shelterbelts and riparian buffers in rural communities. *F. pennsylvanica* is an opportunistic species capable of rapid colonization, as evidenced by its rapid spread in Europe, where it is invasive. High phenotypic plasticity and the Great Plains origin of most of the cultivars in this study may have contributed to the success of this hypothesized process. *F. pennsylvanica* woodlands occupy only 1–4% of the landscape in this region but support a disproportionately large component of biological diversity, including migratory songbirds, gallinaceous birds, and native ungulates [43–46]. Our results indicate that the potential long-term ecological and genetic impacts of unintentional, human-mediated migration of broadly adapted native forest trees merits additional investigation and may not be negative under certain circumstances.

## Implications for conservation of the North American Fraxinus

**Adaptive variation.** Studies of outcrossing, wind pollinated forest trees have shown that adaptive variation occurs at local and regional scales even when connectivity among

populations is high [47, 48]. However, *F. pennsylvanica* provenance tests conducted over the last ninety years provide phenotypic evidence for local adaptation while at the same time indicating that provenance alone does not necessarily predict growth rate, one measure of adaptation. Height in 60 provenances planted at 10 test sites (common gardens) and measured at age six was not consistent with the expectation that provenances closest to a given test site will grow the fastest [49]. Although southern provenances did suffer injury and mortality from cold temperatures in northern test sites, the tallest and the earliest maturing trees at most of the test sites were from southern Ontario and a 'central prairie' region which included provenances from eastern Nebraska, Iowa, and central Illinois. In the Steiner study and in a previous investigation of 13 year old provenances planted in Massachusetts, the northeastern provenances had no growth advantage in test sites closest to northeastern provenances [50]. On the other hand, a provenance test of seedlings from 39 Great Plains provenances from North Dakota, South Dakota, Minnesota, Iowa Nebraska and Kansas did reveal evidence of local adaption for drought tolerance, with provenances in northwest North Dakota being most tolerant [51]. These studies, completed well before the EAB invasion, provide evidence for local adaptation. In the light of the evidence presented here, we hypothesize that the standing genetic variation in *F. pennsylvanica* is impacted by a human-mediated process (i.e. gene flow from landscaping clones), the effect of which cannot be directly disentangled from prior local adaptive variation, however that may be defined, without fine scale functional genomics and intensive phenotyping.

**Conservation of the North American *Fraxinus* under threat from EAB.** Conservation of the gene pools of *F. pennsylvanica*, *F. americana* and the other North American *Fraxinus* under threat from EAB requires consideration of what "gene pool" means given the evidence for locally extensive gene flow from cultivars and porous species boundaries. Gene flow from conspecific susceptible cultivars into susceptible naturally regenerated stands is unlikely to boost defense responses to EAB, and can result in the erosion of local genetic diversity. The effect of admixture between *F. pennsylvanica* and *F. americana* on EAB susceptibility in individual trees has not been closely examined under controlled conditions and merits investigation. Regardless of how the gene pool is defined, interdisciplinary efforts based on forest monitoring, seed collection, long-term breeding programs, landscape genomics and intensive phenotyping will all be required to conserve the existing genetic diversity of the North American *Fraxinus* species and the same time incorporate enough EAB resistance to restore *Fraxinus* populations to the landscape.

## Supporting information

**S1 Fig. *F. pennsylvanica* species density (USA) relative to sampled population locations.** Thirty-eight site locations are labeled after the state in the United States in which they occur. Ten site locations, those in Alberta, Saskatchewan, Manitoba, Ontario, Quebec, New Brunswick, and Nova Scotia, are labeled after the Canadian provinces in which they occur. Species density data shown for the United States only.
(TIF)

**S2 Fig. Genetic groups and admixture inferred for 19 cultivars and 10 *F. americana* comparators.** Panel a) Genetic groups and admixture inferred at *K* = 3. Panel b) Genetic groups and admixture at *K* = 4.
(TIF)

**S1 Table. Provenance and other descriptors for the cultivars included in this study.**
(DOCX)

**S2 Table. Summary statistics and plots for the regression model.** Linear regression model summary statistics using allele richness as the response variable and cultivar incidence as the predictor variable.
(DOCX)

**S3 Table. Microsatellite allele sizes for the entire dataset.**
(XLSX)

## Acknowledgments

We appreciate the assistance of federal, state, provincial, county, township, and municipal governments in the United States and Canada in locating naturally regenerated stands of *F. pennsylvanica* and granting us permission to collect samples. We thank the Minnesota Landscape Arboretum for allowing us access to their *Fraxinus* cultivar collection, and the University of Notre Dame Genomics Core Facility for technical advice and assistance with genotyping.

## Author Contributions

**Conceptualization:** Everett A. Abhainn, Devin L. Shirley, Jennifer L. Koch, Jeanne Romero-Severson.

**Data curation:** Robert K. Stanley, Tatum Scarpato, Jeanne Romero-Severson.

**Formal analysis:** Everett A. Abhainn, Devin L. Shirley, Jeanne Romero-Severson.

**Funding acquisition:** Robert K. Stanley, Jennifer L. Koch, Jeanne Romero-Severson.

**Investigation:** Everett A. Abhainn, Devin L. Shirley, Robert K. Stanley, Jeanne Romero-Severson.

**Methodology:** Everett A. Abhainn, Devin L. Shirley, Robert K. Stanley, Tatum Scarpato, Jennifer L. Koch, Jeanne Romero-Severson.

**Project administration:** Jennifer L. Koch, Jeanne Romero-Severson.

**Resources:** Jeanne Romero-Severson.

**Supervision:** Everett A. Abhainn, Jennifer L. Koch, Jeanne Romero-Severson.

**Validation:** Devin L. Shirley, Robert K. Stanley, Jennifer L. Koch, Jeanne Romero-Severson.

**Visualization:** Everett A. Abhainn, Devin L. Shirley, Robert K. Stanley, Tatum Scarpato, Jeanne Romero-Severson.

**Writing – original draft:** Everett A. Abhainn, Devin L. Shirley.

**Writing – review & editing:** Robert K. Stanley, Jennifer L. Koch, Jeanne Romero-Severson.

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
