## [Decision Letter · Decision Letter 0]

7 Feb 2024

PONE-D-23-37013Gene flow from Fraxinus cultivars into natural stands of Fraxinus pennsylvanica occurs range-wide, is regionally extensive, and is associated with a loss of allele richnessPLOS ONE

Dear Dr. Romero-Severson,

Thank you for submitting your manuscript to PLOS ONE. After careful consideration, we feel that it has merit but does not fully meet PLOS ONE’s publication criteria as it currently stands. Therefore, we invite you to submit a revised version of the manuscript that addresses the points raised during the review process.

We look forward to receiving your revised manuscript.

Kind regards,

Vikas Sharma, Ph.D

Academic Editor

PLOS ONE

Journal Requirements:

"This work was funded by the National Science Foundation grant I0S1025974 ‘Comparative Genomics of Environmental Stress Responses in North American Hardwoods’ (https://www.nsf.gov/div/index.jsp?div=IOS). JR-S also acknowledges support from USDA-USFS APHIS grant 18-IA-11242316-105 (https://www.fs.usda.gov/), USDA-APHIS grant 20-JV-11242303-050 (https://www.aphis.usda.gov/aphis/home/) and The Tree Fund grant 18-JD-01 (https://treefund.org/). RKS acknowledges support from NIH training grant T32GM075762 (https://cbbi.nd.edu/). JLK acknowledges support from USDA APHIS 18-IA-11242316-105, Michigan Invasive Species Grant Program grant IS18-119 (https://www.michigan.gov/invasives/grants/misgp), the Commonwealth of Pennsylvania Department of Conservation and Natural Resources Bureau of Forestry 18-CO-11242316-014 (https://www.dcnr.pa.gov/Pages/default.aspx) and the U.S. Forest Service Special Technology Development Program grant NA-2017-01."

3. Please expand the acronym “USDA-USFS APHIS” (as indicated in your financial disclosure) so that it states the name of your funders in full.

**Additional Editor Comments:**

Authors are advised to go through the comments given by the reviewer and address the issues raised by the reviewer. Overall, the manuscript almost good but needs language revision to make the things/ statements clearer.

Are Genomic SSRs available in Fraxinus?

Reviewers' comments:

Reviewer's Responses to Questions

**Comments to the Author**

1. Is the manuscript technically sound, and do the data support the conclusions?

Reviewer #1: Partly

2. Has the statistical analysis been performed appropriately and rigorously? 

Reviewer #1: Yes

3. Have the authors made all data underlying the findings in their manuscript fully available?

Reviewer #1: No

4. Is the manuscript presented in an intelligible fashion and written in standard English?

Reviewer #1: No

5. Review Comments to the Author

Reviewer #1: This manuscript needed to restructured and written again as its present form create lots of confusions among the reader.

1. In introduction author needs to first explain the Green ash and white ash plant separately with their origin, ecology, uses and function in that region.

2. Authors have used the words green ash, white ash and their scientific names in the whole manuscript carelessly which create confusion among the reader and have to go back in the introduction which one is green ash and which one is white ash. The name should be synchronized and one name should be used in all the manuscript.

3. The last paragraph of introduction line 94-102, author have included the lines of methods, results and conclusion while he should justify why they have conducted such study and what was the hypothesis behind the study?

4. Are cultivar and hybrid same thing? Than use only one terminology.

5. In Method section author should clearly mentioned how many accession he had taken of green ash, suspected cultivar and White ash from 48 populations?

6. Hypothesis have been discussed in the discussion but clear Hypothesis were not given in the introduction section.

7. What is the role of emerald ash borer in depleting the population of Green or White ash? or Cultivar proven to be better resistant form emerald ash borer not discussed.

8. Over all impression of the manuscript is highly confusing with no clear cut objectives and what they achieve by doing the whole study? Do cultivar resulted in depletion of allele richness on Green ash or white ash population? Which species needed to be conserved?

I recommend the author to rewrite the manuscript in sequence from tip to toe so that manuscript can be easily understood to the reader.

6. PLOS authors have the option to publish the peer review history of their article (what does this mean?). If published, this will include your full peer review and any attached files.

Reviewer #1: No

---

## [Author Response · Author response to Decision Letter 0]

16 Mar 2024

Response to the reviewers

Reviewer #1: This manuscript needed to restructured and written again as its present form create lots of confusions among the reader.

We have restructured the abstract and the introduction to clarify what the investigation was about, what the research questions were, and the method we used to address these questions. In the revised abstract, in sentences 1 and 2 we write “Undetected cultivar gene flow, if extensive, could significantly lower genetic diversity within populations, suppress differentiation between populations, generate interspecific admixture not driven by long-standing natural processes, and affect the impact of abiotic and biotic threats. We used 16 EST-SSR markers to genotype 48 naturally regenerated populations of F. pennsylvanica distributed across the native range (1291 trees), 19 F. pennsylvanica cultivars, and one F. americana L. (white ash) cultivar to detect cultivar propagule dispersal into these populations. “

1. In introduction author needs to first explain the Green ash and white ash plant separately with their origin, ecology, uses and function in that region.

The investigation was not focused on the distinctions between F. pennsylvanica and F. americana, as is evident in the title, which we have not changed. We revised the abstract and the introduction to make this clear. Although a result of our investigation was the detection of admixture, this detection does not does not change our main result, the detection of evidence for cultivar gene flow into natural stands and the negative association of cultivar gene flow with allele richness. We included the white ash comparators because we were aware of the existing evidence for the difficulty of discerning the species by morphological and site criteria alone, as stated in the original text. Previous investigations focused on the difficulty of distinguishing these species are cited in the text. Our results pertinent to species admixture are not novel and are not presented as such. Our result does add complexity to the question of what constitutes the F. pennsylvanica gene pool, which we address in the discussion.

2. Authors have used the words green ash, white ash, and their scientific names in the whole manuscript carelessly which create confusion among the reader and have to go back in the introduction which one is green ash and which one is white ash. The name should be synchronized and one name should be used in all the manuscript.

The reviewer is correct. We have revised the text and now use the designations F. pennsylvanica and F. americana consistently throughout the manuscript.

3. The last paragraph of introduction line 94-102, author have included the lines of methods, results, and conclusion while he should justify why they have conducted such study and what was the hypothesis behind the study?

This comment and the reviewer’s first comment indicate that we did not clearly state what the scientific issue was (gene flow from cultivars in natural stands), and what the potential impact of the issue might be (loss of genetic variation). We have restructured the abstract and the introduction to clarify what the investigation was about, what the research questions were, and then state the method we used to address these questions.

4. Are cultivar and hybrid same thing? Than use only one terminology.

The words ‘cultivar’ and ‘hybrid’ are certainly not the same thing. We did not make this clear and carelessly used the word ‘hybrid’ in two places in the text. We have added text explaining what the word ‘cultivar’ means in the introduction and explain that the cultivar designation does not necessarily imply species purity. We deleted the two uses of the word ‘hybrid’ in the original submission. The word ‘hybrid’ no longer appears in the text. Individuals for which we have evidence of interspecific ancestry are now consistently labeled ‘admixed’.

5. In Method section author should clearly mentioned how many accession he had taken of green ash, suspected cultivar, and White ash from 48 population

This comment indicates that we did not make our methods as clear as we could have. In the revised methods section, we clearly define “population”, indicate that all the individuals in the defined population were genotyped under the presumption that they were F. pennsylvanica and explain that the 20 cultivars were from a Fraxinus cultivar collection. We included the passport data for all cultivars in the original submission. In the revised introduction we now state “In this investigation we genotyped 48 naturally regenerated populations of F. pennsylvanica (1291 trees) and 20 cultivars with the same set of 16 EST-SSR markers to detect first-generation progeny from cultivars in the populations we genotyped and assess the impact of such gene flow on population differentiation and population substructure”.

6. Hypothesis have been discussed in the discussion but clear Hypothesis were not given in the introduction section.

We have addressed this in the revisions to the abstract and introduction. In the revised abstract, in the second sentence we now write “. Undetected cultivar gene flow, if extensive, could significantly lower genetic diversity within populations, suppress differentiation between populations, generate interspecific admixture not driven by long-standing natural processes, and affect the impact of abiotic and biotic threats. In this investigation we generated the first range-wide genetic assessment of F. pennsylvanica and included genotypes from 20 commercial cultivars to detect the extent of cultivar gene flow into natural stands.”

7. What is the role of emerald ash borer in depleting the population of Green or White ash? or Cultivar proven to be better resistant form emerald ash borer not discussed.

Population depletion due to the emerald ash borer and possible resistance to the emerald ash borer in the cultivars are outside the scope of this investigation, although we do address the importance of these considerations in the discussion. In the methods, we indicate that we avoided those areas in which EAB has inflicted the most mortality: “We did not collect samples in southern Michigan, northern Indiana, or northern Ohio, as surviving adult trees were unlikely to be representative of the genetic diversity existing before the emerald ash borer invasion”. While the data we have does not address the questions raised by reviewer, the genetic impact of EAB-inflicted mortality and the evaluation of cultivars for response to EAB infestation are important topics for future investigations.

8. Over all impression of the manuscript is highly confusing with no clear cut objectives and what they achieve by doing the whole study? Do cultivar resulted in depletion of allele richness on Green ash or white ash population? Which species needed to be conserved?

We hope that the revisions referenced above clear away the confusion over species distinctions. The methods and the figure provided clearly show that the association between allele richness and cultivar gene flow arises from a linear regression that includes the data from all of the populations, regardless of interspecific admixture. In the discussion we argue that “Conservation of the gene pools of F. pennsylvanica, F. americana and the other North American Fraxinus under threat from EAB requires consideration of what “gene pool” means given the evidence for locally extensive gene flow from cultivars and porous species boundaries.” The data we have supports the expectation of low species boundaries, at least under some circumstances, but this is a matter for separate investigation.

Comments to the Editor and corrections or additions requested by PlosOne

We thank the editor and the reviewer for their time and their comments. We have revised the manuscript guided by the reviewer and editor comments. We have moved the topic of the propagule dispersal capacity of F. pennsylvanica from the introduction, where it may have been distracting, to the discussion. We have not changed the data analyses, the tables, or the figures, other than to change the font in the figures to Ariel.

Corrections or additions

1. Please state what role the funders took in the study. 

2. Please expand the acronym “USDA-USFS APHIS” (as indicated in your financial disclosure) so that it states the name of your funders in full.

JR-S also acknowledges support from the United States Forest Service (https://www.fa.usda.gov) and the Animal and Plant Health Inspection Service( https://www.aphis.usda.gov/aphis/home/) grant 18-IA-11242316-105 .

3. Please include your full ethics statement in the ‘Methods’ section of your manuscript file. In your statement, please include the full name of the IRB or ethics committee who approved or waived your study, as well as whether or not you obtained informed written or verbal consent. If consent was waived for your study, please include this information in your statement as well

We have included an ethics statement in the Methods.

Response to editor question

1. Are Genomic SSRs available in Fraxinus?

In the text we now reference studies done with genomic microsatellites from F. excelsior and one study done with eight microsatellites from F. pennsylvanica and F. excelsior. We used none of these, as none met our criteria of reliable amplification and polymorphism in both F. pennsylvanica and F. americana. We developed our own set of EST-SSR that did meet these criteria and reference the paper in which we described our screening methods.

---

## [Decision Letter · Decision Letter 1]

1 Apr 2024

Gene flow from Fraxinus cultivars into natural stands of *Fraxinus pennsylvanica* occurs range-wide, is regionally extensive, and is associated with a loss of allele richness

PONE-D-23-37013R1

Dear Dr. Romero-Severson,

We’re pleased to inform you that your manuscript has been judged scientifically suitable for publication and will be formally accepted for publication once it meets all outstanding technical requirements.

Kind regards,

Vikas Sharma, Ph.D

Academic Editor

PLOS ONE

Additional Editor Comments (optional):

Authors have addressed all the raised issues in a nice manner and modified the manuscript as per the suggestions. It is a good study and now can be accepted for publication.

Please check line no.103 "ecosystems types"

Please check line no. 290 F. americana and F. americana admixture (heading) if it is right?

Reviewers' comments:

Reviewer's Responses to Questions

**Comments to the Author**

1. If the authors have adequately addressed your comments raised in a previous round of review and you feel that this manuscript is now acceptable for publication, you may indicate that here to bypass the “Comments to the Author” section, enter your conflict of interest statement in the “Confidential to Editor” section, and submit your "Accept" recommendation.

Reviewer #1: All comments have been addressed

2. Is the manuscript technically sound, and do the data support the conclusions?

Reviewer #1: Yes

3. Has the statistical analysis been performed appropriately and rigorously? 

Reviewer #1: Yes

4. Have the authors made all data underlying the findings in their manuscript fully available?

Reviewer #1: Yes

5. Is the manuscript presented in an intelligible fashion and written in standard English?

Reviewer #1: Yes

6. Review Comments to the Author

Reviewer #1: Author have improved and synchronize the manuscript to be easily understandable and removed the confusion. They have addressed all the comments raised in this manuscript clearly. I recommend this manuscript to be accepted and published in your journal.

7. PLOS authors have the option to publish the peer review history of their article (what does this mean?). If published, this will include your full peer review and any attached files.

Reviewer #1: **Yes: **Dr. Vikrant Jaryan

---

## [Editor Report · Acceptance letter]

26 Apr 2024

PONE-D-23-37013R1 

PLOS ONE

Dear Dr. Romero-Severson, 

I'm pleased to inform you that your manuscript has been deemed suitable for publication in PLOS ONE. Congratulations! Your manuscript is now being handed over to our production team.

Kind regards, 

on behalf of

Dr. Vikas Sharma 

Academic Editor

PLOS ONE